# The Effect of Noise Exposure on High-Frequency Hearing Loss among Chinese Workers: A Meta-Analysis

**DOI:** 10.3390/healthcare11081079

**Published:** 2023-04-10

**Authors:** Ping Yang, Hui Xie, Yajing Li, Ke Jin

**Affiliations:** 1Faculty of Architecture and Urban Planning, Chongqing University, Chongqing 400045, China; 2Key Laboratory of New Technology for Construction of Cities in Mountain Area, Chongqing 400045, China

**Keywords:** China, occupational noise exposure, high frequency hearing loss, meta-analysis

## Abstract

The purpose of this study was to determine the burden of high frequency noise-induced hearing loss (HFNIHL) in Chinese workers exposed to hazardous noise through meta-analysis, to evaluate the major risk factors of HFNIHL in Chinese workers, and to provide evidence for reducing the risk of HFNIHL. We searched for relevant studies on HFNIHL published between January 1990 and June 2022. Inclusion and exclusion criteria were established to screen the literature, and the quality of the studies was assessed. Meta-analysis was performed using the software Stata 17.0. A total of 39 studies involving 50,526 workers in different industries were included in this study. The incidence of HFNIHL in the noise-exposed group (36.6%) was higher than that in the control group (12.5%), with a pooled odds ratio (OR) of 5.16 and a 95% confidence interval (CI) of 4.10–6.49. Sensitivity analysis showed that the results of this meta-analysis were stable. Funnel plots and Egger’s test showed no publication bias. Subgroup analysis showed heterogeneity among the results of different studies, which was related to gender, publication year, age, length of work, and type of industry. The dose–response analysis showed that cumulative noise exposure (CNE) and length of work were the main risk factors for HFNIHL. This study suggests that the detection rate of HFNIHL is high in Chinese workers, the risk of HFNIHL increases rapidly when CNE reaches 90 dB(A)·year, and the first 15 years of exposure to noise is a period of increasing risk. Therefore, reasonable measures for preventing hearing loss should be taken to reduce the risk of occupational HFNIHL.

## 1. Introduction

The burden of disease and death caused by environmental pollution is becoming a public health challenge, especially in developing countries worldwide. Prior to the 1990s, many countries focused on scientific research and policy-making on pollutants such as air pollution, water pollution, and solid waste. In fact, as early as 1971, a working group of the World Health Organization (WHO) concluded that noise was a major threat to human well-being [1]. More and more evidence shows that noise pollution, similar to other pollutants, has a wide range of negative impacts on human health, social harmony, and economic development. Therefore, noise is an important public health problem in modern society and can lead to hearing loss, sleep disturbance, cardiovascular disease, social dysfunction, decreased productivity, impaired teaching and learning, absenteeism, increased accidents, and other problems [2].

Hearing loss is defined as a reduced sensitivity to normal sounds. According to the “World Report on Hearing” published by the WHO in 2021, currently more than 1.5 billion people (nearly 20% of the global population) suffer from hearing loss, and it is estimated that more than 700 million people may have disabled hearing loss by 2050 [3]. Hearing loss can be caused by damage to any part of the peripheral and central auditory system. The main causes of sensorineural hearing loss are aging, noise exposure, and exposure to therapeutic drugs with ototoxic side effects [4]. Noise-induced hearing loss (NIHL) is hearing damage caused by sudden exposure to pulse noise or long-term exposure to high-intensity noise [5]. Hearing loss due to workplace noise exposure has been a serious health problem. The prevalence of occupational noise exposure-induced hearing loss in the United States is around 10% [6]. The prevalence of NIHL was 51.3% in plywood factory workers and 90.3% in steel mill workers in India [7,8]. The prevalence of NIHL among migrant workers in Kuwait was 20.4% [9]. Globally, 16% of adult disabling hearing loss is attributed to occupational noise, and this proportion varies from 7% to 21% in different regions [10]. The World Health Organization estimates that billions of people worldwide continue to face unavoidable risks of NIHL due to exposure to high noise levels [11].

Currently, China is in the stage of accelerating industrialization, with traditional industries and new equipment manufacturing industries expanding and industrial noise increasing, especially the noise generated by machine operations such as air compressors, ventilation fans, textile machines, and metal processing machines. Factors such as a lack of ideal noise reduction designs in factories, lack of protection engineering in workshops, weak awareness of personal noise control and prevention, and prolonged working hours may lead to workers being exposed to hazardous noise environments for a long time. Due to the high incidence of high-frequency hearing loss (HFNIHL) in workers exposed to noise, the purpose of this study is to determine the occupational burden of HFNIHL in Chinese workers exposed to harmful noise levels and to evaluate the main risk factors leading to HFNIHL in Chinese workers, providing a basis for reducing the risk of HFNIHL.

The impact of industrial modernization on workers’ hearing loss in China has re-ceived attention. A recent meta-analysis of high-frequency hearing loss in Chinese workers based primarily on the literature from 2017 to 2019 was conducted by Kuang et al. [12]. The results showed that the detection rate of high-frequency hearing loss due to occupational noise exposure among Chinese workers was 9.45%. However, in the study by Kuang et al. their literature inclusion criteria did not set a non-noise exposed control group, thus their findings may have underestimated the prevalence of high frequency hearing loss. In addition, their meta-analysis study suffered from a low number of literature inclusion, lack of risk factor analysis, and high publication bias in the results.

To objectively understand the burden of occupational noise exposure hearing loss in Chinese workers, this study searched the literature of high-frequency hearing loss studies from January 1990 to June 2022, developed reasonable literature inclusion criteria, conducted a dose–response analysis of risk factors for high-frequency hearing loss, and performed sensitivity tests and publication bias tests on the results of meta-analysis.

## 2. Materials and Methods

### 2.1. Literature Search

The search was conducted on both Chinese and English databases, including the China National Knowledge Infrastructure (CNKI), Wanfang Database, Chinese Biomedical Literature Database (CBM), Web of Science, PubMed, and MEDLINE. The search terms “industrial noise”, “occupational noise”, “workers”, “hearing loss”, and “China” were used to search the published literature from January 1990 to June 2022.

### 2.2. Inclusion and Exclusion Criteria

Inclusion criteria were: (1) studies in both Chinese and English that investigated hearing loss due to occupational noise exposure in China, (2) participants had no other underlying diseases, (3) studies that reported the number of participants exposed to different noise levels and the number of participants with high-frequency noise-induced hearing loss (HFNIHL), and (4) studies that included a control group with a different length of occupational exposure. Exclusion criteria were: (1) studies of noise exposure in workers outside of China, (2) studies of hearing loss or deafness that were not related to noise exposure, (3) studies of non-occupational noise exposure, (4) studies that did not include a noise background control group, (5) animal studies on noise-induced hearing loss (NIHL), (6) studies with incomplete data on the number or prevalence of NIHL cases, (7) studies with evident errors or incomplete data on the demographic characteristics of participants, and (8) books, conference proceedings, and news reports on noise exposure.

### 2.3. Data Extraction and Quality Assessment

Data extraction included the first author, publication year, sample size, basic demographic information (age, gender, length of occupational exposure), industry type, equivalent sound levels (LA_eq_), cumulative noise exposure (CNE), and the prevalence of high-frequency noise-induced hearing loss (HFNIHL).

Two independent reviewers evaluated the quality of the included studies using the Joanna Briggs Institute Meta Analysis of Statistics Assessment and Review Instrument (JBI-MAStARI) [13], which included eight items such as sample definition, background description, and hearing loss measurement, etc. Each item was scored as “yes”, “no”, “unclear”, or “not applicable”. Studies with a score of 5 or more were considered high-quality studies, while those with a score less than 5 were considered low-quality studies. In cases of disagreement between the reviewers, a third reviewer was consulted.

### 2.4. Statistical Analysis

Meta-analysis was conducted using the software Stata 17.0. The effect size was measured using the odds ratio (OR), and a 95% confidence interval (95% CI) was reported. Heterogeneity was assessed using I^2^. When I^2^ ≤ 50%, a fixed-effect model was used for analysis, and when I^2^ > 50%, a random-effect model was used. Subgroup and sensitivity analyses were performed based on the characteristics of the included studies. Publication bias was assessed using funnel plots and Egger’s regression test.

## 3. Results

### 3.1. Literature Search

A total of 704 articles were retrieved from the database search. After screening (Figure 1), 39 studies were included in the meta-analysis. All 39 studies were cross-sectional in design. The overall quality of the literature was high, as each included study had a score greater than five (Table 1).

### 3.2. Characteristics of Included Studies

The basic characteristics of the 39 included articles are shown in Table 1. The total sample size was 50,526 workers, including 38,075 workers in the noise exposure group and 12,451 workers in the control group. The studies involved various industries, such as the chemical industry, power plants, machinery manufacturing, mining, steel, food processing, tobacco, wood processing, and drilling, and all of them used a cross-sectional design.

### 3.3. Meta-Analysis of the Effect of Noise Exposure on High-Frequency Hearing Loss

A heterogeneity test showed that there was heterogeneity among the studies (I^2^ = 89.3%, *p* < 0.001) (Figure 2). Therefore, a random effects model was used for analysis. The statistical results in Table 2 showed that the average noise exposure levels of workers in the noise exposure group and the control group were 103.6 ± 5.8 dB(A) and 65.2 ± 6.4 dB(A), respectively, and the incidence of HFNIHL in the noise exposure group (36.6%) was higher than that in the control group (12.5%). The forest plot (Figure 2) showed that there was a correlation between noise exposure and the risk of HFNIHL in workers (OR = 5.16, 95% CI: 4.10–6.49), which was statistically significant (*p* < 0.001).

### 3.4. Subgroup Analysis

The 39 studies included in this meta-analysis were divided into five subgroups based on publication year, gender, mean age, length of employment, and industrial type to explore the contribution of these factors to the heterogeneity among the study results. The results are shown in Table 3. In addition, the forest plots of the association of different subgroups with HFNIHL are shown in Appendix A.

Publication year subgroup: The pooled OR value for studies published before 2010 was 7.4 (95% CI: 5.43–10.09), with an I^2^ of 61.8% and *p* < 0.05. The pooled OR value for studies published after 2010 was 4.29 (95% CI: 3.26–5.64), with an I^2^ of 90.6% and *p* < 0.001.

Gender subgroup: The pooled OR value for male workers was 12.22 (95% CI: 5.22–28.61), with an I^2^ of 95% and *p* < 0.001. The pooled OR value for female workers was 7.89 (95% CI: 3.56–17.48), with an I^2^ of 68.4% and *p* < 0.05. The gender composition of the subjects was not specified in 31 studies.

Age subgroup: The pooled OR value for workers under 40 years of age was 4.65 (95% CI: 3.90–5.55), with an I^2^ of 68.9% and *p* < 0.001. The pooled OR value for workers over 40 years of age was 5.88 (95% CI: 3.21–10.76), with an I^2^ of 95.2% and *p* < 0.001. One study did not specify the age of the subjects.

Length of employment subgroup: The pooled OR value for workers with less than 10 years of employment was 4.88 (95% CI: 3.63–6.56), with an I^2^ of 83.5% and *p* < 0.001. The pooled OR value for workers with more than 10 years of employment was 5.42 (95% CI: 3.63–8.08), with an I^2^ of 90.2% and *p* < 0.001. Three studies did not specify the length of employment of the subjects.

Industrial type subgroup: The pooled OR value for the manufacturing industry was 4.62 (95% CI: 3.87–5.51), with an I^2^ of 64.7% and *p* < 0.001. The pooled OR value for the mining industry was 2.29 (95% CI: 0.70–7.58), with an I^2^ of 95.8% and *p* < 0.001. Only two studies involved noise exposure in the mining industry, and the heterogeneity was very significant. There was only one study each in the transportation industry and other industries, and they had no statistical significance.

It can be seen that heterogeneity in the results of different studies correlated with gender, year of publication, age, duration of employment, and type of industry.

### 3.5. Dose–Response Analysis

The dose–response relationship between cumulative noise exposure (CNE) and length of work with the risk of HFNIHL is a nonlinear curve, which was fitted using a spline function. The dose–response curve between CNE and the risk of HFNIHL is shown in Figure 3a, which reveals an intersection of linear and nonlinear curves with a turning point in the nonlinear curve at a CNE of 90 dB(A)·year. This indicates that workers have a relatively low risk of high-frequency hearing loss in the initial phase of noise exposure, and the risk increases rapidly as the exposure level reaches a CNE of 90 dB(A)·year.

The dose–response curve between length of work and the risk of HFNIHL is shown in Figure 3b, which also shows an intersection of linear and nonlinear curves with a turning point at 15 years of work. Figure 3b demonstrates that workers are exposed to a greater risk of HFNIHL in the first 15 years of noise exposure, and the risk tends to plateau after 15 years of exposure.

Gender was a categorical variable, and age was divided into categorical variables by <30 and ≥30 years. From Table 4, it can be seen that there is a positive correlation between male gender and the risk of HFNIHL (combined OR = 2.83, 95% CI: 1.36–5.49). Furthermore, an increase in age (≥30 years) is also positively correlated with the risk of HFNIHL (combined OR = 2.33, 95% CI: 1.62–3.36).

### 3.6. Sensitivity Analysis and Publication Bias

Due to the inclusion of a large number of studies in this research, to evaluate the stability of the meta-analysis model, as shown in Figure 4, each study was systematically removed one by one, and the pooled OR value and 95% CI remained almost unchanged (OR = 5.15, 95% CI: 4.10–6.47), indicating that the results of this study are reliable and robust.

The funnel plot (Figure 5) constructed based on the incidence rate of HFNIHL is, basically, symmetrical, indicating the absence of publication bias in the included literature. In addition, the Egger linear regression test was used to quantitatively evaluate the potential publication bias. As shown in Figure 6, there is still some heterogeneity among the included literature, but the calculation results indicate that there is no significant publication bias (t = 1.98, *p* > 0.05).

## 4. Discussion

This study reviewed and analyzed the literature on high-frequency hearing loss (HFNIHL) caused by occupational noise in China from 1990 to 2022. The meta-analysis results showed that the incidence rate of HFNIHL caused by occupational noise exposure among Chinese workers was 36.6%. Basu et al. conducted a meta-analysis of occupational noise-induced hearing loss in India and found an incidence rate of 50% for NIHL in Indian workers [53]. Mrena et al. found that the incidence rate of occupational NIHL in Finland showed a decreasing trend from 19.2% in 1999 to 8.3% in 2002 [54].

Noise-induced hearing loss (NIHL) initially presents as sensorineural hearing loss in higher frequency ranges (3 kHz–6 kHz), which is known as HFNIHL. As time goes on, hearing loss can gradually develop in lower frequency ranges [55]. In this study, we only included the incidence of HFNIHL, while some of the literature included in this study provided the incidence of NIHL, which includes the number of people with HFNIHL and speech frequency hearing loss (SFNIHL) [26,29]. Therefore, the incidence rate of occupational HFNIHL in China is 36.6%, but the incidence rate of occupational NIHL should be higher than 36.6%. It can be seen that the incidence rate of occupational NIHL in China is similar to that of India and is at a moderate to high level.

Noise-induced hearing loss is irreversible, and there is currently a lack of effective treatments. Therefore, prevention measures are particularly important to reduce the incidence of NIHL. Measures such as strengthening the awareness of noise hazards among enterprise management, setting up necessary noise protection facilities, controlling noise propagation, implementing shift work systems for noise job positions, reducing exposure time in noisy environments, strengthening occupational health education for noise workers, and strictly wearing personal protective equipment are recommended [36].

There was a correlation between noise intensity and high-frequency noise-induced hearing loss (HFNIHL). When studying hearing loss in a boiler factory, Ni et al. found that the detection rates of HFNIHL among workers exposed to noise intensities of 86–90 dB(A) and 101–105 dB(A) were 33% and 78%, respectively [16]. Hu et al. investigated the hearing loss in a steel pipe factory, and the detection rate of workers’ HFNIHL was 60% when the noise intensity was ≤95 dB(A) and 70% when the noise intensity was ≤105 dB(A) [23]. To explore whether there was a dose–response relationship between noise intensity and the risk of HFNIHL, in this study, the relationship between different noise intensities and the risk of HFNIHL was conducted using cumulative noise exposure (CNE) as a dose indicator. Previous individual studies have all suggested that there was a dose–response relationship between noise exposure intensity and the risk of high-frequency hearing loss. However, due to the small sample sizes of each study, it was not enough to form a dose–response relationship based on a large-scale, large-sample, and multi-industry study. This study collected a large number of studies, conducted screening and merging, and concluded, through meta-analysis, that workers will be at higher risk of HFNIHL when they are exposed to a CNE of 90 dB(A)·year. This is of great significance for guiding hearing protection practices.

The occupational risk of HFNIHL in workers is related to their length of work, with a higher detection rate of HFNIHL being associated with longer exposure times to noise [27,35]. Liang et al. believed that noise had a cumulative effect on hearing loss over time [56]. Length of work is an indicator that can easily measure exposure time to noise, but there are few studies based on large samples that explore the dose–response relationship between length of work and the risk of HFNIHL. The dose–response curve of this study shows that there is a turning point in the curve at 15 years of service, and the curve rises gradually after 15 years. This suggests that workers have a higher risk of developing HFNIHL in the first 15 years of noise exposure and that the risk remains but is relatively low after 15 years of work. Nevertheless, some studies suggest that workers’ risk of HFNIHL from noise exposure is mainly concentrated in the first 10 years [57]. Whether our conclusion is closer to the truth still requires further research. Noise-induced hearing loss is generally believed to be caused by damage to the cochlear hair cells, and there is currently no direct evidence that biological damage to the human cochlear hair cells caused by noise accumulates over time [58]. However, in order to reduce the risk of HFNIHL, monitoring the temporary threshold shift (TTS) of workers exposed to high noise levels, especially those with less than 15 years of service, is valuable for preventing HFNIHL.

The study included different literature with heterogeneity in the results. Pan et al. reported in 2009 that the detection rate of HFNIHL in workers exposed to noise intensity of 85–100 dB(A) was 69.1% [20], while Zhou et al. reported in 2017 that the detection rate of HFNIHL in workers exposed to noise intensity of 85–105 dB(A) was 28% [34]. It can be seen that the difference in the detection rate of HFNIHL is large due to the difference in publication year. As shown in Table 1, under the same conditions of age, work duration, gender, and industry type, there are still differences in the detection rate of HFNIHL among different studies. The conclusion drawn from this subgroup analysis is that the factors of publication year, gender, mean age, work duration, and industry type all have an impact on the heterogeneity of different research results. Therefore, it is suggested that more literature should be included and more reasonable inclusion criteria and grouping rules should be developed in order to obtain more rigorous analytical results in the future.

Due to the heterogeneity of the studies included in this research, a sensitivity analysis was performed using the one-by-one exclusion method to examine the original meta-analysis results. The results showed that the pooled OR value and 95% CI hardly changed, indicating that the literature screening method, inclusion criteria, and analysis model selection in this study were qualified. Therefore, the conclusions drawn from this research were relatively robust. Funnel plots were constructed, and the Egger linear regression test was used for a qualitative and quantitative evaluation of publication bias, indicating that the conclusions of this study were credible and that publication bias is not present.

Hearing protection devices (HPDs) are an effective means of preventing NIHL for workers in the workplace. In 1980, China established a noise standard for industrial workplaces, which states that noise in the workplace must not exceed 85 dB(A). All workers exposed to noise levels above 90 dB(A) are required to wear hearing protection devices (HPDs). Some of the literature in this study reported that HPDs were provided in factories, but only a few workers were willing to wear them [14,19]. The main reason for some workers to avoid using ear protection was the lack of ergonomic comfort. In addition, wearing HPDs does not reduce noise annoyance due to workers’ hearing threshold shifts that cause them to adapt to the high noise level environment in the workplace [46,49]. Therefore, there is still much work to be done to incorporate the wearing of hearing protection devices as a hearing conservation measure. For example, workers need to be educated about the role of HPDs in reducing the risk of high-frequency hearing loss, and workers must adhere to a strict regime for wearing HPDs when exposed to noise.

A systematic review and meta-analysis of Chinese occupational NIHL has been conducted previously by another author [12,59]. However, this study was not a simple repetition of another author’s work. Firstly, the time span for collecting literature in this study was different, and a relatively large number of studies were included in this meta-analysis. Secondly, in this study, two dose–response relationship curves were established separately, revealing that CNE and length of work were the main risk factors for HFNIHL. Finally, qualitative and quantitative tests for publication bias were performed. Therefore, the conclusions drawn in this study were more reliable and robust.

There are, admittedly, some limitations to this study. All of the included literature are cross-sectional studies, and there is a lack of large sample cohort studies, which may affect the causal relationship judgment of HFNIHL risk. Conference papers, in-house hearing test records for different industrial sectors, and studies in languages other than Chinese and English were not included, which may introduce some bias. In some included studies, the frequency selection for measuring hearing thresholds was not entirely consistent. We hope to conduct some rigorously designed cohort studies of noise exposure and NIHL risk in large samples for different industrial sectors in the future. This will help us further understand the natural patterns of NIHL occurrence and development.

## 5. Conclusions

The prevalence of HFNIHL in Chinese workers is relatively high (36.6%), and cumulative noise exposure and length of employment are the main risk factors for HFNIHL. In the early stage of noise exposure, the risk of HFNIHL increases slowly, but after reaching the sensitive point of CNE around 95 dB(A)·years, the risk increases rapidly. The first 15 years of noise exposure is a period of increased risk for HFNIHL, after which the risk tends to flatten out.

In order to reduce the incidence of occupational HFNIHL, engineering control measures should first be used to reduce noise generation and minimize occupational noise exposure. Secondly, a comprehensive workplace hearing loss prevention plan should be developed, such as noise assessment, noise control, worker hearing monitoring, and appropriate use of hearing protectors. Finally, sound occupational noise control regulations should be established, and regulatory compliance should be checked and monitored by relevant regulatory authorities. This will greatly reduce the burden of occupational noise-induced hearing loss.

## Figures and Tables

**Figure 1 healthcare-11-01079-f001:**
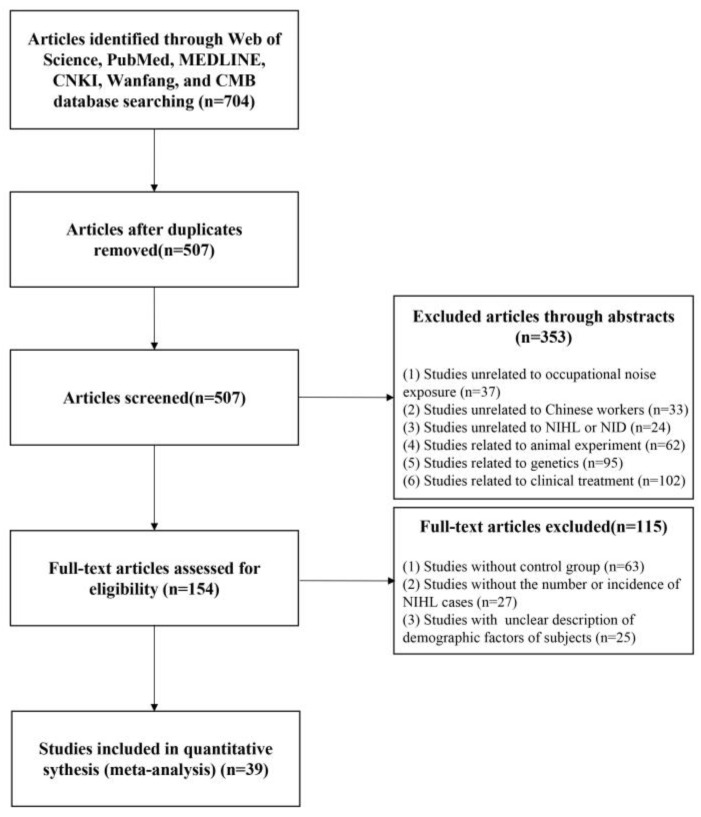
Flow chart of literature screening.

**Figure 2 healthcare-11-01079-f002:**
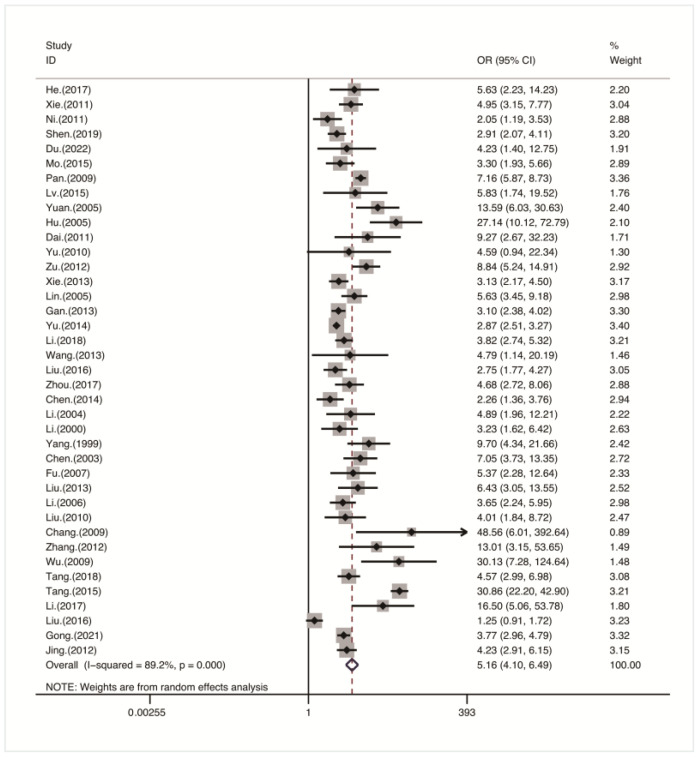
Forest plot of noise exposure and HFNIHL risk [14,15,16,17,18,19,20,21,22,23,24,25,26,27,28,29,30,31,32,33,34,35,36,37,38,39,40,41,42,43,44,45,46,47,48,49,50,51,52].

**Figure 3 healthcare-11-01079-f003:**
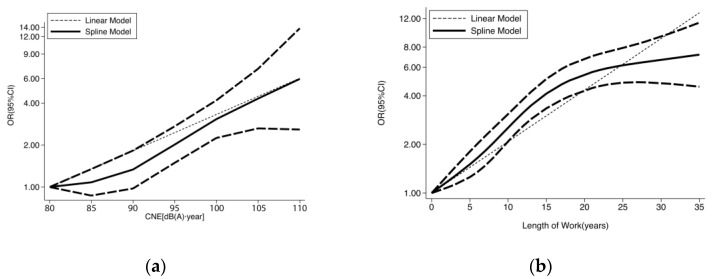
(**a**) Dose–response relationships between CNE and HFNIHL; (**b**) Dose–response relationships between length of work and HFNIHL.

**Figure 4 healthcare-11-01079-f004:**
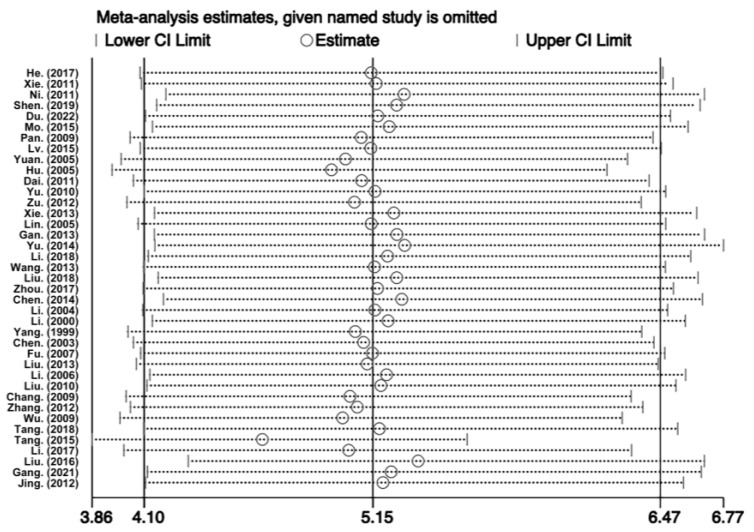
Sensitivity analysis of noise exposure and HFNIHL [14,15,16,17,18,19,20,21,22,23,24,25,26,27,28,29,30,31,32,33,34,35,36,37,38,39,40,41,42,43,44,45,46,47,48,49,50,51,52].

**Figure 5 healthcare-11-01079-f005:**
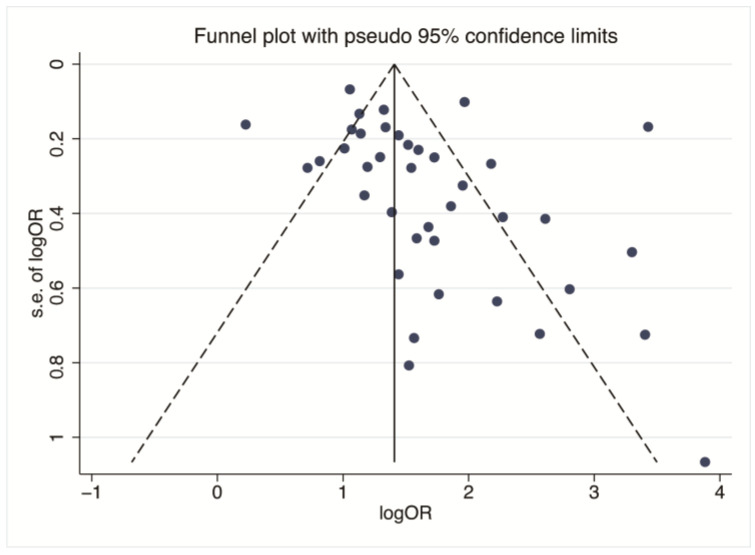
Funnel plot analysis of noise exposure and risk of HFNIHL.

**Figure 6 healthcare-11-01079-f006:**
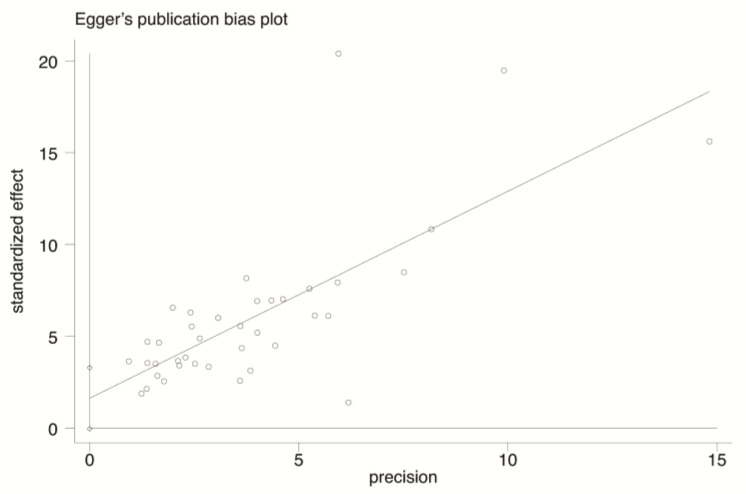
Egger linear regression test of noise exposure and risk of HFNIHL.

**Table 1 healthcare-11-01079-t001:** Main characteristics of 39 included studies on noise exposure and HFNIHL risk.

First Author and Year	Sample Size	Type of Industry	Male/%	Age/Years	Length of Work/Years	LA_eq_/dB(A)	CNE/dB(A)·Year	HFNIHL/%	SFNIHL/%	Quality Assessment
He, 2017 [14]	531	Chemical plant	95.7	33.52 ± 8.39	9.20 ± 6.68	67.57–90.65	-	21.1	-	8
Xie, 2011 [15]	495	Hardware, printing, etc.	-	28.9	6.3	90.0	-	52.6	-	7
Ni, 2011 [16]	214	Boiler factory	90.2	42.9 ± 8.5	17.6 ± 11.9	77.0–123.8	85–105	58.1	8.6	8
Shen, 2019 [17]	1100	Toy factory, mining, etc.	70.0	39.1 ± 4.1	0.5–10	82.5	-	14.1	1.8	7
Du, 2022 [18]	158	Woodworking	63.9	37.24 ± 5.33	4.25 ± 0.79	85–99	-	29.2	-	7
Mo, 2015 [19]	1288	Industrial Park	60.7	32.32 ± 4.27	13.73 ± 10.18	89.7	-	9.0	-	7
Pan, 2009 [20]	2000	Ship	100	20–60	-	110.0	85–100	69.1	10.9	7
Lv, 2015 [21]	246	Wood processing	40.9	36.0	4.3	68.2–104	-	25.9	-	7
Yuan, 2005 [22]	174	Hot forging workshop	100	36.9 ± 9.0	19.7 ± 8.2	87–109	-	36.2	13.8	8
Hu, 2005 [23]	191	Steel pipe mills	-	33.6 ± 4.2	12.7 ± 3.0	87–109	95–110	68.3	35.5	7
Dai, 2011 [24]	159	Hydropower Station	76.1	35.4 ± 7.0	14.2 ± 6.6	60–106	103.5 ± 3.1	17.6	5.7	6
Yu, 2010 [25]	110	PLA Kitchen	-	-	-	80.0	-	10.0	7.3	5
Zu, 2012 [26]	778	Mechanical processing	58.0	28.2 ± 6.1	2.9 ± 3.3	65.9–96.6	92.1 ± 4.4	31.9	-	8
Xie, 2013 [27]	2127	Papermaking Enterprise	99.2	31.21 ± 4.82	9.51 ± 4.74	86.1	-	22.6	12.3	6
Lin, 2005 [28]	1000	Machinery, etc.	56.8	19–51	<5–>15	80.5–104.5	-	12.0	1.3	5
Gan, 2013 [29]	3182	Chemical companies	87.0	33.5 ± 8.4	-	-	-	6.5	4.2	5
Yu, 2014 [30]	14,843	-	88.6	28.4 ± 7.0	4.0 ± 3.5	60.7–109.0	-	52.0	7.6	6
Li, 2018 [31]	5661	Machinery, automobile	96.0	33.9 ± 9.4	0–25	89.6 ± 7.5	-	15.7	-	8
Wang, 2013 [32]	441	Gem process	42.6	40.2 ± 9.5	11.1 ± 6.3	62.9–102.3	-	14.17	3.4	8
Liu, 2016 [33]	1659	Power Generation Enterprise	63.4	42.36 ± 7.95	18.5 ± 11.18	78.1	-	11.5	7.0	7
Zhou, 2017 [34]	594	Steel Enterprise	0	38.9 ± 8.2	14.8 ± 8.9	-	85–105	28.0	-	7
Chen, 2014 [35]	1273	Power Plant	74.0	44.5 ± 6.8	21.5 ± 8.3	86.9 ± 12.9	-	14.0	-	7
Li, 2004 [36]	129	Boiler-Manufacturing	94.6	32.04 ± 9.61	4.52 ± 2.97	82–108	-	51.2	-	7
Li, 2000 [37]	300	Power Branch Plant, etc.	-	34.13 ± 9.97	10.5 ± 6.2	103.4	80–105	24.7	-	6
Yang, 1999 [38]	154	Metal processing plant	87.0	32.6 ± 7.7	8.6 ± 7.3	125.0	-	30.5	-	5
Chen, 2003 [39]	482	-	0	23.13 ± 5.77	7.0	93–103	-	16.8	24.9	5
Fu, 2007 [40]	207	Chemical Plant	67.1	19–50	0.2–18	85–89	-	36.2	12.1	8
Liu, 2013 [41]	594	Machinery processing	87.9	36.2	11.7	71.5–106.4	85–100	22.0	-	7
Li, 2006 [42]	1050	Artificial Gem Plant	96.5	24.0 ± 3.9	2.7 ± 2.1	89.2 ± 2.8	-	31.0	-	8
Liu, 2010 [43]	710	Tobacco company	54.6	36.8 ± 7.9	17.3 ± 9.4	42.4–90.9	-	20.9	2.3	7
Chang, 2009 [44]	75	Liquefied Petroleum Gas Cylinder	100.0	42.4 ± 7.9	10.0 ± 7.4	49–98	-	29.3	-	8
Zhang, 2012 [45]	645	Electronic Technology	75.0	26.3 ± 3.6	5.0 ± 3.4	86.6 ± 2.6	-	11.8	-	8
Wu, 2009 [46]	600	Shoe factory	0	20–51	1–16	71–96	-	9.8	1.7	5
Tang, 2018 [47]	1346	Automobile, etc.	-	34.7 ± 8.7	20.0 ± 9.4	88.3 ± 16.1	-	8.0	0.1	7
Tang, 2015 [48]	2200	-	100	22–55	0–25	85.59 ± 1.92	-	33.3	-	7
Li, 2017 [49]	445	-	77.8	19–66	6.4 ± 5.3	78.5–93.9	-	11.9	-	6
Liu, 2016 [50]	738	Coal miners	66.9	43.14 ± 6.66	>10	60–110	-	28.9	10.0	8
Gong, 2021 [51]	1653	Auto parts factory	70.0	37.0	8.0	87.0	94–100	40.0	-	8
Jing, 2012 [52]	974	Oilfield drilling	100	41.57 ± 9.12	15.88 ± 10.96	60.8–105.7	107.93 ± 6.45	18.4	2.0	8

Note: “-”, indicates that the literature does not mention the content.

**Table 2 healthcare-11-01079-t002:** Characteristics of the noise exposed group compared to the control group.

Group	Sample Size	Age/Years	Length of Work/Years	LA_eq_/dB(A)	HFNIHL/%	SFNIHL/%	Pooled OR	95% CI
Noise	38,075	35.4 ± 6.4	10.8 ± 6.4	103.6 ± 5.8	36.6	12.1	5.16	4.10–6.49
Control	12,451	36.2 ± 8.7	12.4 ± 9.8	65.2 ± 6.4	12.5	6.2	-	-

**Table 3 healthcare-11-01079-t003:** Subgroup analysis of noise exposure and risk of HFNIHL.

Subgroup	Number of Studies	Pooled OR	95% CI	*p*	I^2^/%
**Publication Year**					
<2010	13	7.4	5.43–10.09	0.002	61.8
≥2010	26	4.29	3.26–5.64	<0.001	90.6
**Gender**					
Male	5	12.22	5.22–28.61	<0.001	95
Female	3	7.89	3.56–17.48	0.042	68.4
Both	26	3.82	3.18–4.60	<0.001	73.8
NA	5	5.75	3.31–9.98	0.011	69.4
**Age (Mean)** **/years**					
<40	25	4.65	3.90–5.55	<0.001	68.9
≥40	13	5.88	3.21–10.76	<0.001	95.2
NA	1	4.59	0.94–22.34	-	-
**Length of Work (Mean)/years**					
<10	17	4.88	3.63–6.56	<0.001	83.5
≥10	19	5.42	3.63–8.08	<0.001	90.2
NA	3	4.71	2.26–9.82	<0.001	92.1
**Type of Industry**					
Manufacturing	31	4.62	3.87–5.51	<0.001	64.7
Mining	2	2.29	0.70–7.58	<0.001	95.8
Transportation	1	7.16	5.87–8.72	-	-
Other	1	4.59	0.94–22.34	-	-
NA	4	9.9	2.19–44.86	<0.001	89.2

Note: NA indicates not available.

**Table 4 healthcare-11-01079-t004:** Gender, age, length of work, cumulative noise exposure and risk of HFNIHL.

Factor	Number of Studies	Group	Pooled OR	95% CI
Gender	4	Female	-	-
		Male	2.83	1.46–5.49
Age/years	3	<30	-	-
		≥30	2.33	1.62–3.36
Length of Work/years	15	<5	-	-
		5–10	1.51	1.26–1.81
		10–15	3.15	2.61–3.81
		15–20	4.70	3.77–5.86
		20–25	5.93	4.69–7.48
		25–30	6.48	4.84–8.66
		≥30	7.18	4.55–11.34
CNE/dB(A)·year	8	<85	-	-
		85–90	1.08	0.87–1.35
		90–95	1.34	0.98–1.83
		95–100	2.02	1.49–2.73
		100–105	3.07	2.25–4.19
		105–110	4.31	2.64–7.06
		≥110	5.97	2.59–13.77

## Data Availability

Not applicable.

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
