# Peer review of "The Effect of Noise Exposure on High-Frequency Hearing Loss among Chinese Workers: A Meta-Analysis"

_healthcare, 2023, doi:10.3390/healthcare11081079_

Round 1
Reviewer 1 Report
Introduction: Although largely analysed and well known, the topic of the study is actual and interesting.
Due to the many papers on this argument already published, it could be interesting to highlight both limitations of the previous studies and novelties of this meta-analysis (as also reported in the end of the paper too) in chapter “Introduction”.
Aim: the aim of the study is well explained, although it could be also introduced by comparisons between previous studies (see above).
Methods: correctly used. More details about search query results should be added to “Supplemental materials”.
Results: well reported.
Discussion: well conducted
Conclusion: well reported. Update of literature, highlights of limitation of previous studies and emerged novelties are important as the actuality of the topic.
Suggestions: It would be interesting to known the relationship between Hl and use of personal protective device, if used.
Author Response
Point 1: Due to the many papers on this argument already published, it could be interesting to highlight both limitations of the previous studies and novelties of this meta-analysis (as also reported in the end of the paper too) in chapter “Introduction”.
Response 1: We think this is an excellent suggestion. Changes have been made as suggested, see lines 68-81.
Point 2: Aim: the aim of the study is well explained, although it could be also introduced by comparisons between previous studies (see above).
Response 2: Changes have been made as suggested, see lines 68-81.
Point 3: Methods: correctly used. More details about search query results should be added to “Supplemental materials”.
Response 3: Figure 1 was updated. The new figure 1 contained more details about search query results.
Point 4: Results: well reported.
Response 4: Sincere thanks.
Point 5: Discussion: well conducted
Response 5: Sincere thanks.
Point 6: Conclusion: well reported. Update of literature, highlights of limitation of previous studies and emerged novelties are important as the actuality of the topic.
Response 6: Sincere thanks.
Point 7: Suggestions: It would be interesting to known the relationship between Hl and use of personal protective device, if used.
Response 7: Changes have been made as suggested, see lines 304-316.

Reviewer 2 Report
authors described the influence a long exposure by high frequency noise to hearing loss incidence by workers in China.
The paper is an metaanalyse.
The basic question is if the exposure by noise were detected that no any workers used the saving of hearing function?
That the exposed persons were in the areas with high risk of noise places, or the places were not under the special procedures with saving of hearing?
Exist any special procedures for saving of persons under noise influence ( rules, low a.o.).
Is necessary to do any special rules for the workers in this areas or not, or is possibele to recommand it.
Author Response
Point 1: The basic question is if the exposure by noise were detected that no any workers used the saving of hearing function?
Response 1: We think this is an excellent suggestion. Changes have been made as suggested, see lines 304-316.
Point 2: That the exposed persons were in the areas with high risk of noise places, or the places were not under the special procedures with saving of hearing?
Response 2: Changes have been made as suggested, see lines 304-316.
Point 3: Exist any special procedures for saving of persons under noise influence ( rules, low a.o.).
Response 3: We have made changes to the discussion, see lines 304-316.
Point 4: Is necessary to do any special rules for the workers in this areas or not, or is possibele to recommand it.
Response 4: Yes. we think that workers need to be educated about the role of HPDs in reducing the risk of high-frequency hearing loss, and workers must adhere to a strict regime for wearing HPDs when exposed to noise. Please see lines 304-316 of revison.
